# Gastro-Esophageal Cancer: Can Radiomic Parameters from Baseline ^18^F-FDG-PET/CT Predict the Development of Distant Metastatic Disease?

**DOI:** 10.3390/diagnostics14111205

**Published:** 2024-06-06

**Authors:** Ricarda Hinzpeter, Seyed Ali Mirshahvalad, Roshini Kulanthaivelu, Andres Kohan, Claudia Ortega, Ur Metser, Amy Liu, Adam Farag, Elena Elimova, Rebecca K. S. Wong, Jonathan Yeung, Raymond Woo-Jun Jang, Patrick Veit-Haibach

**Affiliations:** 1University Medical Imaging Toronto, Toronto Joint Department Medical Imaging, University Health Network, Sinai Health System, Women’s College Hospital, University of Toronto, Toronto, ON M5G 2N2, Canada; ricarda.stolzmann-hinzpeter@usz.ch (R.H.); roshini.kulanthaivelu@queensu.ca (R.K.); andres.kohan@uhn.ca (A.K.); claudia.ortega@uhn.ca (C.O.); ur.metser@uhn.ca (U.M.); adam.farag@uhn.ca (A.F.); patrick.veit-haibach@uhn.ca (P.V.-H.); 2Institute for Diagnostic and Interventional Radiology, University Hospital Zurich, 8091 Zurich, Switzerland; 3Department of Biostatistics, Princess Margaret Cancer Centre, University Health Network, University of Toronto, Toronto, ON M5G 1X6, Canada; zhihuiamy.liu@uhn.ca; 4Department of Medical Oncology, Princess Margaret Cancer Centre, University Health Network, University of Toronto, Toronto, ON M5G 2C4, Canada; elena.elimova@uhn.ca; 5Department of Radiation Oncology, Princess Margaret Cancer Centre, University Health Network, University of Toronto, Toronto, ON M5G 2C4, Canada; rebecca.wong@uhn.ca (R.K.S.W.); raymond.jang@uhn.ca (R.W.-J.J.); 6Division of Thoracic Surgery, Department of Surgery, Toronto General Hospital, University Health Network, University of Toronto, Toronto, ON M5G 2C4, Canada; jonathan.yeung@uhn.ca

**Keywords:** positron emission tomography, computed tomography, gastroesophageal cancer, radiomics, metastasis, survival

## Abstract

We aimed to determine if clinical parameters and radiomics combined with sarcopenia status derived from baseline ^18^F-FDG-PET/CT could predict developing metastatic disease and overall survival (OS) in gastroesophageal cancer (GEC). Patients referred for primary staging who underwent ^18^F-FDG-PET/CT from 2008 to 2019 were evaluated retrospectively. Overall, 243 GEC patients (mean age = 64) were enrolled. Clinical, histopathology, and sarcopenia data were obtained, and primary tumor radiomics features were extracted. For classification (early-stage vs. advanced disease), the association of the studied parameters was evaluated. Various clinical and radiomics models were developed and assessed. Accuracy and area under the curve (AUC) were calculated. For OS prediction, univariable and multivariable Cox analyses were performed. The best model included PET/CT radiomics features, clinical data, and sarcopenia score (accuracy = 80%; AUC = 88%). For OS prediction, various clinical, CT, and PET features entered the multivariable analysis. Three clinical factors (advanced disease, age ≥ 70 and ECOG ≥ 2), along with one CT-derived and one PET-derived radiomics feature, retained their significance. Overall, ^18^F-FDG PET/CT radiomics seems to have a potential added value in identifying GEC patients with advanced disease and may enhance the performance of baseline clinical parameters. These features may also have a prognostic value for OS, improving the decision-making for GEC patients.

## 1. Introduction

Esophageal and gastroesophageal junction malignancies are among the leading causes of cancer morbidity and mortality worldwide and responsible for a significant disease burden globally [1]. At initial presentation, a significant proportion of the patients are found to have advanced metastatic disease [2]. Thus, a significant proportion of patients are excluded from curative surgical treatment and will be delivered palliative therapy instead, while for patients with early-stage disease, surgery (commonly along with neoadjuvant chemotherapy or chemoradiotherapy) may be a curative option [1,3].

Positron emission tomography/computed tomography (PET/CT) with ^18^F-fluorodeoxyglucose (^18^F-FDG) is a cornerstone in patient staging and a complementary technique to endoscopic ultrasound (EUS), especially to determine advanced metastatic disease [1,4]. ^18^F-FDG-PET/CT semi-quantitative metabolic parameters have been shown to have both diagnostic and prognostic value [5,6]. In addition, it has been reported that the value of ^18^F-FDG-PET/CT can be potentially enhanced even further using a quantitative image evaluation approach such as radiomics [7,8,9]. Radiomics refers to extracting and analyzing large amounts of quantitative features on medical images, which can be utilized to feed artificial intelligence methods, such as machine learning algorithms [10]. Its general workflow includes image reconstruction, segmentation, feature extraction, feature selection, and data analysis.

Several clinical parameters were reported to be of value in gastroesophageal cancer prognostication, such as disease stage, histopathology grade, sex, age (particularly ≥70 years), body mass index (BMI), and functional impairment [3,11,12,13]. Also, it has been found that patients with more aggressive gastroesophageal malignancies are often malnourished and cachectic due to cancer-related symptoms. To assess this, sarcopenia, a severe depletion of skeletal muscle mass, has been introduced as a marker [14,15]. Sarcopenia score is recognized as a prognostic factor and most accurately and reproducibly measured as skeletal muscle area on CT images [3,14].

Thus, we aimed to assess the value of ^18^F-FDG-PET/CT-derived radiomics features in predicting the development of metastatic disease by classifying gastroesophageal cancer patients into early-stage and advanced metastatic disease and comparing these two groups. We included various clinical factors and sarcopenia measurements to reach a combined model with a more robust prediction of patient status to evaluate the added value that ^18^F-FDG-PET/CT-derived radiomics may add to the established clinical routine. As a secondary goal, we evaluated these factors to assess their prognostic value in predicting overall survival (OS).

## 2. Materials and Methods

### 2.1. Study Cohort

In this retrospective IRB-approved study, all patients who were referred to our institution for esophageal or gastroesophageal cancer initial staging between November 2008 and December 2019 were gathered from the institutional registry. Patients who underwent ^18^F-FDG-PET/CT as part of their cancer staging at our center were enrolled in the study. Clinical and histopathology data were obtained from the institutional database, including demographic data (age, sex, race, BMI, and ECOG score at clinical presentation), tumor characteristics, histopathology information (subtype and grade), disease stage (early-stage disease with locoregional involvement versus advanced disease with distant metastasis; M0 vs. M1), treatment details, and follow-up data.

### 2.2. ^18^F-FDG-PET/CT Acquisition and Image Analysis

^18^F-FDG-PET/CT imaging was performed based on the standardized institutional protocol in our center. PET scans were acquired in 3D mode with two dedicated in-line PET/CT scanners (two generations of the Biograph scanner); Siemens Biograph mCT 40 (Siemens Healthineers, Erlangen, Germany) and a Biograph PET/CT scanner (Siemens Healthineers, Erlangen, Germany). Both scanners were from the same vendor, and acquisition and reconstruction parameters were harmonized (EARL-compliant) to minimize differences in image reconstruction and uptake values. Subjects were instructed to fast for a minimum duration of 6 h prior to the radiopharmaceutical administration. Then, ^18^F-FDG was delivered intravenously for 4–5 MBq/kg of body weight, with a maximum dose of 550 MBq. Additionally, for gastrointestinal tract opacification, iodinated oral contrast material was administered (no intravenous iodinated contrast administration). A spiral CT scan was conducted from the skull base to the upper thighs. The scan was performed with the following parameters: 120 kV peak, 40 to 105 mAs, 3.0 mm slice width, 2.0 mm collimation, an overlap of 2.0 mm, 0.8 s rotation time, and 8.4 mm feed/rotation. Following the CT completion, PET images were acquired approximately 60 min after ^18^F-FDG intravenous administration, for a duration of 3 min/bed position, with each patient undergoing 5–9 bed positions based on their body height. A PET scan using scatter correction was obtained, covering the identical transverse field of view of the CT acquisition (skull base to upper thighs). The PET image dimensions were pixel size 2.6 × 2.6 mm^2^ and a slice thickness of 3.27 mm, filtered with a 4 mm full width at half-maximum Gaussian filter.

^18^F-FDG-PET/CT interpretation was conducted by expert radiologists using an imaging workstation (Mirada XD Workstation, Mirada Medical, Ltd.; Oxford, UK), and their reports (diagnosing early-stage versus advanced disease) were extracted from the institutional registry. Two radiologists and one nuclear medicine specialist experienced in oncologic imaging re-read the images and measured the mean, max, and peak standard uptake values (SUVs), as well as SUVs normalized by lean body mass (SUL), for the primary tumor. SUVs and SULs were calculated with a semi-automatically drawn volume of interest (VOI) covering the entire tumor, as defined by PET images on multiple slices encompassing the entire lesion. The SUVmean was evaluated in a similar way and was derived from the whole-tumor VOI. The SUVpeak and SULpeak were measured using a pre-defined 1 cm^3^ spherical VOI centered on the pixel with the highest uptake.

Radiomic features were extracted using commercially available and Image Biomarker Standardization Initiative (IBSI)-compliant open-source software (LIFEx, version 6.3, Inserm, Paris, France; lifexsoft.org [16]). Tumor segmentation was performed by two radiologists/nuclear medicine physicians. The primary tumor was segmented manually on the CT images in a slice-by-slice manner. On PET images, it was contoured semi-automatically using a thresholding-based approach, applying three different thresholds on the defined VOI, including whole-tumor (using the relative contrast with the background normal tissue uptake), as well as 40% and 70% of the primary tumor’s SUVmax thresholds, as previously described [3]. The radiomics features included 307 different features and were extracted from the segmented volumes in accordance with the IBSI guidelines [17]. They contained conventional metrics features reporting the mean, median, maximum, and minimum values of the voxel intensities on the image-, size-, and shape-based histogram features, such as volume; compacity and sphericity, including their asymmetry (skewness); flatness (kurtosis); uniformity and randomness; and textural features (e.g., GLCM (Gray-Level Co-Occurrence Matrix), GLRLM (Grey-Level Run Length Matrix), NGLDM (Neighborhood Grey-Level Different Matrix), and GLZLM (Grey-Level Zone Length Matrix)). Note that all commonly used PET/CT-derived parameters, including SUVs, MTV, and TLG, as well as Hounsfield units (HU), were included in the extracted features.

Moreover, sarcopenia measurements were performed and calculated, as discussed previously [15]. In summary, they were calculated from the CT component of ^18^F-FDG-PET/CT at the lumbar spine L3 level. For skeletal muscle identification, HU was used (threshold −29 to 150 HU). Then, the total skeletal muscle area in cm^2^ was calculated using Slice-O-Matic software (version 5.0; TomoVision, Magog, QC, Canada). Skeletal muscle index was measured by normalizing the muscle area (cm^2^) for the patient’s height (m^2^). To interpret, keletal muscle index cut-offs for sarcopenia definition were 45.4 cm^2^/m^2^ in males and 34.4 cm^2^/m^2^ in females [14].

### 2.3. Statistical Analysis

Continuous and categorical variables were presented as mean (±standard deviation) and frequency (with percentage), respectively. For metastatic status prediction (early-stage M0 vs. advanced metastatic M1 disease), the association of the studied parameters was evaluated using the Chi-square test or Student’s *t*-test for the categorical or continuous variables, respectively. The significantly different (two-sided *p*-value < 0.01) factors between the early-stage and advanced cohorts were considered to enter the model building after removing all except one of the highly correlated variables in each modality. Additionally, variables with >10% missing data were excluded.

For model building, a k-folded Light Gradient Boosting Machine (LGBM) classifier was utilized. Appendix A shows this machine learning algorithm’s pipeline. It is noteworthy to provide the rationale behind why we opted for this method of modelling. In summary, we chose a high-fold LGBM classifier to successfully handle our high-dimensional complex radiomics dataset, capture non-linear relationships between features, obtain a more stable estimate of the models’ true performance, and, finally, bolster the confidence in the reliability of our results. LGBM is an efficient learning algorithm that has shown excellent accuracy in classification tasks using radiomics data, e.g., exhibiting the most outstanding differential performance when compared to other classifiers [18,19]. To develop the baseline clinical model, the included variables were age, sex, race, BMI, ECOG score, and histopathology subtype and grade. The single-modality (CT and PET) radiomics models were then built. Next, the single-modality models were combined with each other to form hybrid PET/CT radiomics models and also merged with the baseline clinical model and sarcopenia score to find the best combination of variables for patient classification. Model performance was quantified and visualized considering accuracy and calculated using the internal k-fold cross-validation method. Also, the area under the curve (AUC), recall, precision, and F1 score (harmonic mean of the precision and recall) were calculated for each model. DeLong’s test was used to compare the performances (AUCs) of the built models pairwise.

The best combination (highest performance metrics overall) contained PET/CT-derived radiomics features, along with clinical data and the sarcopenia score. This model was tuned using 1000 iterations to reach the final best model. Its characteristics were bagging fraction = 0.9, bagging frequency = 3, boosting type = ‘gbdt’, colsample by tree = 1.0, feature fraction = 0.5, importance type = ‘split’, learning rate = 0.1, max depth = −1, min child samples = 11, min child weight = 0.001, min split gain = 0.9, estimators number = 210, jobs number = −1, leaves number = 8, random state = 42, regression alpha = 1 × 10^−6^, regression lambda = 0.15, silent = ‘warn’, subsample = 1.0, subsample for bin = 200,000, and subsample frequency = 0. This pipeline was also followed using a random forest classifier, and its results are provided in the Appendix A.

For the prediction of OS, patient survival was calculated from the date of ^18^F-FDG PET/CT acquisition to either the date of a death report issuance or the last date of follow-up. Univariable Cox regression analysis was performed, and parameters found to be significant (*p*-value < 0.05) were considered for multivariable Cox regression analysis, with hazard ratio and 95% confidence interval (95% CI) calculated. Again, parameters with a high Pearson’s correlation (cut-off ≥ 0.7) were considered for removal.

The continuous variables that were significantly correlated with the response were converted to categorical variables. For this purpose, a receiver operating characteristic (ROC) curve was drawn for each variable, and the best cut-off was defined based on the Youden index. The stepwise method was utilized for the variable selection.

Lastly, for illustration, Kaplan–Meier survival curves were drawn for the significant radiomics features from the multivariable Cox analysis. All data were analyzed using Statistical Package for the Social Sciences (SPSS software; version 26, IBM, Chicago, IL, USA). Unless otherwise specified, a two-tailed *p*-value < 0.05 was considered statistically significant.

## 3. Results

In this study, 243 patients (mean age = 64 years; male percentage = 79%) with esophageal or gastroesophageal cancer were enrolled. Of them, 115 (47%) had early-stage disease, and 128 (53%) were found to be advanced. Detailed characteristics of the studied patients can be seen in Table 1.

Radiomics features were extracted from all primary tumors, the details of which can be found in Appendix A. In summary, 34 CT and 70 PET features differed significantly (*p*-values < 0.01) between early-stage and advanced cohorts. Among the PET-derived features, 30, 21, 13, and 6 were related to the whole-tumor, 40% SUVmax, 70% SUVmax, and SUVpeak thresholding method, respectively. Classification models were built using these features.

The LGBM classifier’s detailed results are shown in Table 2. The baseline clinical model (including age, sex, race, BMI, ECOG score, and histopathology subtype and grade) had an accuracy and AUC of 64% and 70%, respectively. By pairwise comparison, the baseline clinical model had a significantly lower AUC than the combination of CT-radiomics + clinical + sarcopenia, PET-radiomics + clinical + sarcopenia, PET/CT-radiomics + clinical, and PET/CT-radiomics + clinical + sarcopenia (non-directional *p*-values < 0.05). Among all built models, PET-only radiomics had the lowest overall performance (accuracy and AUC of 51% and 54%, respectively), while CT radiomics had significantly higher single-modality overall performance (accuracy and AUC of 72% and 78%, respectively; *p*-value < 0.001).

Our final model (highest performance metrics overall) included hybrid/combined PET/CT radiomics features, along with clinical data and integrated sarcopenia score. Note that, when comparing AUCs statistically, we found that our final model was not significantly better than some other built models. However, considering all performance metrics and incorporating the knowledge from the literature, we chose the mentioned combination as the final proposed model. After using 1000 iterations to have a more robust cross-validation and get closer to the true performance of the final model by decreasing variance, our best model revealed accuracy, AUC, recall, precision, and F1-score of 80%, 88%, 84%, 80%, and 82%, respectively (Figure 1; Table 2). Detailed results (models’ performance and feature importance) of the LGBM classifier, as well as the Random Forest classifier, can be found in the Appendix A.

Considering the survival analysis, at the median follow-up of 13 months (average = 22 months), 182 (75%) patients were deceased. Several features showed significant OS predictions (*p*-values < 0.05) in the univariable Cox analysis, including six clinical, four CT, and twenty-two PET (nine whole-tumor, nine 40% SUVmax, and four 70% SUVmax thresholding) features. Details can be found in Appendix A. After removing highly correlated variables, five clinical, four CT, and five PET (two whole-tumor, two 40% SUVmax, and one 70% SUVmax thresholding) features remained to enter the multivariable analysis (shown in Table 3).

In the multivariable Cox analysis of the studied features to predict OS (Table 4), among the clinical factors, three retained their significance, namely having advanced disease, age ≥ 70 years, and an ECOG score ≥ 2. Considering radiomics features, only one CT-derived and one PET-derived (whole-tumor threshold) features were significant in the multivariable analysis. The Kaplan–Meier curves of the significant radiomics features are displayed in Figure 2.

## 4. Discussion

In this study, our findings showed that ^18^F-FDG-PET/CT-derived radiomics features of the primary tumor, along with sarcopenia score and combined clinical factors, might have the potential to identify gastroesophageal cancer patients with advanced metastatic disease. Our developed classification model, including these parameters, reached an accuracy and AUC of 80% and 88%, respectively. Also, radiomics features showed benefits in patients’ OS prognostication and could improve survival prediction by adding value to patients’ baseline clinical information. In the multivariable analysis, CT- and PET-derived features, although not revealing high hazard ratios, could improve OS prediction of the most significant studied clinical factors, including having advanced disease, age ≥ 70 years, and an ECOG score ≥ 2.

There are some published studies in the literature regarding the value of ^18^F-FDG-PET/CT radiomics features in esophageal and gastroesophageal junction cancer patients. To our knowledge, investigations on the role of PET/CT-derived radiomics in predicting metastatic disease and classifying patients into early (M0) versus advanced (M1) disease are, however, scarce, making our study novel in this regard. Jayaprakasam et al. performed a study to predict T2 vs. T3/4 and N0 vs. N1/2 disease (not M0 vs. M1 as we assessed) using PET/CT-derived radiomics [9]. Although they did not include clinical parameters in their model building, there were some interesting findings. For T-stage classification, their single-modality (CT-only and PET-only) models had lower diagnostic accuracies than the dual-modality (PET/CT) model (70.4%, 70.4%, and 81.5%, respectively). They concluded that their developed model had the potential even to surpass the accuracy of EUS. For nodal status classification, PET-only and PET/CT models showed higher accuracies than the CT-only radiomics model (86.2%, 86.2%, and 69.0%, respectively). In another study, Zhang et al. used pre-treatment PET-derived radiomics of the primary tumor to predict esophageal cancer (only adenocarcinoma subtype) lymph node metastasis [20]. They showed that the combination of radiomics features with clinical parameters provided the best results (AUC = 82%). However, as a strength of their study, they performed external validation and revealed that their results were only partly externally replicated (AUC = 69%).

More similar to our outcome, Wu et al. studied single-modality CT-derived (not PET/CT) radiomics, along with clinical parameters, to classify patients into stages I-II versus III-IV [21]. They reported CT radiomics AUCs of 79% and 76% for their primary and validation cohorts, respectively. However, they did not add clinical parameters to assess the possible additive value of radiomics features. In a later study, Zhu et al. aimed to predict gastroesophageal cancer patients’ distant metastasis status similar to our study [22]. However, similar to Wu et al., for their radiomics evaluation, they utilized single-modality CT features (not PET/CT) only. Their baseline clinical model, including age, histopathology grade, and N stage, reached an AUC of 73%. Comparing their various developed models, we see that the combination of radiomics features with clinical parameters (AUC= 83%) had better diagnostic performance than the baseline clinical model and radiomics-only model, thus supporting our study findings.

Considering survival prognostication, there are several studies available on ^18^F-FDG-PET/CT in the literature. Foley et al. studied single-modality ^18^F-FDG PET-derived textural information, along with the clinical parameters, in a large cohort [23]. Like us, they showed that both disease stage and age are significant prognostic factors for OS. In addition, they found that the treatment choice of patients was predictive, which can be partly viewed as similar to our classification of the disease (early-stage vs. advanced disease) since they binarized therapies into curative and palliative. Among ^18^F-FDG-PET/CT-derived quantitative variables, three entered their final model (TLG logarithm, histogram energy logarithm, and histogram kurtosis) and, like our results, had additive prognostic value to the baseline clinical model. Similarly, in the before-mentioned study by Zhang et al., the combination of radiomics and clinical models provided the best OS prognostication, supporting the potential added value of radiomics assessment [20].

Additionally, in a recent investigation, Amrane et al. studied the prognostic value of pre-therapeutic single-modality ^18^F-FDG PET radiomics to predict OS in gastroesophageal junction cancer patients (*n* = 97) [7]. Notably, like our study, they utilized LIFEx software for feature extraction. Their studied clinical parameters included age, sex, histological subtype, human epidermal growth factor receptor-2 expression, metastatic status, and initial pathologic stage, which were mostly similar to the parameters we included in our study. In their multivariable Cox analysis, significant factors were surgical treatment, SUVmean, and histogram entropy. Notably, their dichotomization cut-off for age was 65 years, rendering it an insignificant parameter in the multivariable analysis. Furthermore, in another similar study, Hinzpeter et al. recently investigated the ability of ^18^F-FDG-PET/CT-derived radiomics to predict patient OS [3]. However, in that study, the included population was limited to inoperable advanced metastatic disease. Like the current study, the ECOG score was a significant predictor of OS. It was also found that, among distant metastases, osseous metastasis was highly correlated with worse OS. Additionally, one CT-derived radiomics feature (NGLDM Coarseness) and one PET-derived feature (GLZLM SZLGE) were found to be predictive, which were not the same as the significant features found in the current paper’s mixed population. Additionally, age was found not to be a significant predictor. However, age was also not significant as a continuous variable in our study and only showed its significance after being binarized, highlighting the age grouping value. Also noteworthy, contrary to our study, it was shown that, among the advanced-stage patients, the sarcopenia score was a significant factor in OS prediction in the multivariable analysis.

This study had several limitations to mention. First, there are inherent drawbacks due to the retrospective nature of the study, as most, if not all, of the artificial intelligence-based and radiomics studies are retrospective, making our results at least comparable to the current literature. Second, we had a monocentric patient population and did not test our models on an external validation cohort, thus limiting our results’ generalizability, though we evaluated a comparably larger number of patients, and a robust internal validation was obtained. This issue is also applicable to the OS prognostication findings. Third, for OS prediction, we did not include different therapeutic approaches to the patients and limited our patient characteristics to their information at the time of pre-operative ^18^F-FDG-PET/CT imaging, at which time point they were evaluated as limited vs. advanced stage. We also did not assess radiomics changes following therapy to evaluate delta radiomics measurements’ additive value. Fourth, the follow-up period was somehow limited and may affect our OS assessment, although patients with advanced metastatic disease accounted for nearly half of our patient population. Lastly, although we did not find any significant outlier in our database which could be attributed to the timeline of patient recruitment, there is ample evidence in the literature that imaging protocol variations (e.g., acquisition parameters and post-process variables) can have an impact on radiomics studies [24,25,26,27]. Thus, since our time span for retrospective investigation was from 2008 to 2019, the variations in the imaging protocol over time might have an impact on our data extraction, which, again, underscores the values of a prospective harmonized investigation and external validation.

In conclusion, ^18^F-FDG-PET/CT-derived radiomics features may have the potential to identify gastroesophageal cancer patients with advanced metastatic disease as compared to those with locoregional limited disease, providing additive value to clinical factors and sarcopenia. ^18^F-FDG-PET/CT radiomics may also enhance patients’ OS prediction by adding value to patients’ baseline clinical information. For future studies and to address any concerns about our monocentric study design, performing an external validation is highly recommended.

## Figures and Tables

**Figure 1 diagnostics-14-01205-f001:**
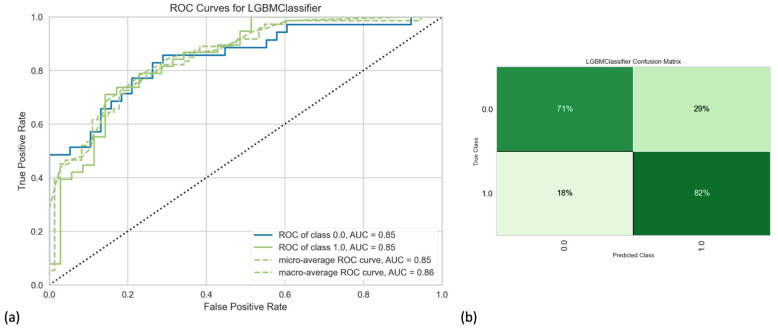
(**a**) Receiver operating characteristic curve and (**b**) confusion matrix of the final best model developed by Light Gradient Boosting Machine (LGBM) classifier.

**Figure 2 diagnostics-14-01205-f002:**
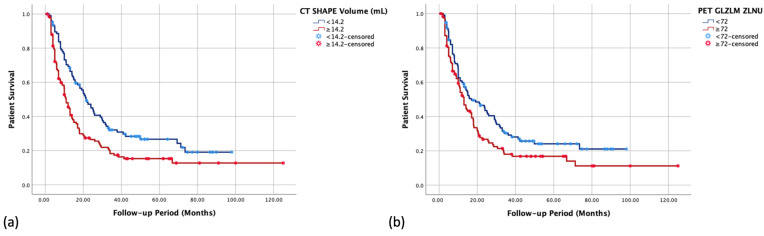
Kaplan–Meier survival curves of the two significant radiomics features from the multivariable Cox analysis to predict overall survival: (**a**) CT-derived and (**b**) PET-derived.

**Table 1 diagnostics-14-01205-t001:** Detailed characteristics of the studied patients (total = 243) in each cohort and comparison between them (non-metastatic vs. metastatic disease).

	Non-Metastatic (*n* = 115)	Metastatic (*n* = 128)	*p*-Value
**Age**	64.8 ± 11.1	63.5 ± 11.7	0.399
**Age ≥ 70 (%)**	40 (35%)	43 (34%)	0.572
**Sex [male] (%)**	90 (78%)	102 (80%)	0.875
**Race [Asian] (%)**	4 (4%)	11 (9%)	0.115
**ECOG ≥ 2 (%)**	8 (7%)	27 (21%)	0.001 *
**BMI**	28.1 ± 5.6	24.4 ± 4.9	<0.001 *
**Sarcopenia score**	48.4 ± 8.7	43.2 ± 9.7	<0.001 *
**Sarcopenic patients (%)**	29 (25%)	60 (47%)	<0.001 *
**Histology [adenocarcinoma] (%)**	112 (97%)	82 (64%)	<0.001 *
- **Grade (%):**			-
- **Gx**	21 (18%)	30 (23%)	
- **G1**	6 (5%)	5 (4%)	
- **G2**	39 (34%)	42 (33%)	
- **G3**	49 (43%)	51 (40%)	
**Grade [G3] (%)**	49 (43%)	51 (40%)	0.696
- **T Stage (%):**			-
- **Tx**	29 (25%)	83 (65%)	
- **T0**	0 (0%)	1 (1%)	
- **Tis**	1 (1%)	0 (0%)	
- **T1**	6 (5%)	1 (1%)	
- **T2**	13 (11%)	2 (2%)	
- **T3**	65 (57%)	33 (26%)	
- **T4**	1 (1%)	8 (6%)	
**T Stage ≥ T3 (%)**	66 (57%)	41 (32%)	<0.001 *
**Primary tumor SUVmax**	12.9 ± 7.8	15.4 ± 7.8	0.011 *
**Primary tumor SUVmean**	8.5 ± 4.6	8.4 ± 3.8	0.816
**Primary tumor SUVpeak**	10.7 ± 6.9	12.9 ± 6.6	0.012 *
**Primary tumor SULmax**	9.1 ± 5.6	11.5 ± 5.8	0.001 *
**Primary tumor SULmean**	6.1 ± 3.3	6.3 ± 2.8	0.627
**Primary tumor SULpeak**	7.9 ± 5.0	9.7 ± 4.9	0.005 *

* Statistically significant.

**Table 2 diagnostics-14-01205-t002:** Light Gradient Boosting Machine (LGBM) classifier results.

	Accuracy (%)	AUC (%)	Recall (%)	Precision (%)	F1 Score (%)
**Baseline clinical model ***	64	70	60	68	63
**CT radiomics**	72	78	77	73	74
**PET radiomics**	51	54	57	54	55
**PET/CT radiomics**	67	78	71	69	34
**CT radiomics + clinical data**	76	84	78	77	77
**CT radiomics + sarcopenia score**	68	77	72	69	70
**CT radiomics + clinical + sarcopenia score**	77	83	79	79	78
**PET radiomics + clinical data**	69	77	67	72	68
**PET radiomics + sarcopenia score**	68	69	72	69	70
**PET radiomics + clinical + sarcopenia score**	74	80	74	76	75
**PET/CT radiomics + clinical data**	76	85	80	77	78
**PET/CT radiomics + sarcopenia score**	71	78	76	72	73
**PET/CT radiomics + clinical + sarcopenia ****	79	85	80	80	80
**Finalized highly iterated cross-validated model**	80	88	84	80	82

* Included age, sex, race, BMI, ECOG, histopathology, and grade. ** Best combination of parameters—used for the finalized model.

**Table 3 diagnostics-14-01205-t003:** Multivariable analysis’ significant features in overall survival prediction.

Modality	Parameter (Defined Cut-Off)	Hazard Ratio (95% CI)	*p*-Value
Clinical (*n* = 5)	Having advanced disease	2.606 (1.921–3.535)	<0.001
	Age ≥ 70 years	1.545 (1.136–2.102)	0.006
	ECOG ≥ 2	3.403 (2.286–5.066)	<0.001
	Being Sarcopenic	1.871 (1.385–2.526)	<0.001
	Having SCC/undifferentiated pathology	1.580 (1.102–2.265)	0.013
CT (*n* = 4)	CT SHAPE volume (mL) (14.2)	1.006 (1.002–1.010)	0.002
	CT SHAPE sphericity (0.545)	0.064 (0.015–0.270)	<0.001
	CT NGLDM contrast (0.05)	0.006 (0.000–0.389)	0.016
	CT GLZLM GLNU (67)	1.001 (1.000–1.002)	0.009
PET (whole tumor; *n* = 2)	SHAPE sphericity (0.743)	0.089 (0.024–0.338)	<0.001
	GLZLM ZLNU (72)	1.001 (1.001–1.002)	<0.001
PET (40% SUVmax; *n* = 2)	GLRLM SRLGE (0.0018)	0.001 (0.001–0.082)	0.044
	NGLDM coarseness (0.11)	0.001 (0.001–0.041)	0.015
PET (70% SUVmax; *n* = 1)	SHAPE volume (Voxel) (100)	1.002 (1.001–1.003)	<0.001

**Table 4 diagnostics-14-01205-t004:** Significant parameters in the multivariable Cox analysis for overall survival prediction (*p*-values < 0.05).

Modality	Parameter	Hazard Ratio (95% CI)	*p*-Value
Clinical (*n* = 3)	Having advanced disease (reference: no)	2.566 (1.849–3.561)	<0.001
	Age ≥ 70 years (reference: no)	1.538 (1.099–2.153)	0.012
	ECOG ≥ 2 (reference: no)	2.799 (1.794–4.366)	<0.001
CT (*n* = 1)	CT SHAPE volume (mL) (reference: <14.2)	1.043 (1.012–1.074)	0.005
PET (whole tumor; *n* = 1)	GLZLM ZLNU (reference: <72)	1.001 (1.001–1.002)	0.001

## Data Availability

The original contributions presented in the study are included in the article/Appendix A; further inquiries can be directed to the corresponding author/s.

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
