# Peer review of "Gastro-Esophageal Cancer: Can Radiomic Parameters from Baseline ^18^F-FDG-PET/CT Predict the Development of Distant Metastatic Disease?"

_diagnostics, 2024, doi:10.3390/diagnostics14111205_

Round 1

Reviewer 1 Report

Comments and Suggestions for Authors

Journal: Diagnostics

Manuscript Title: Gastro-esophageal cancer: Can radiomic parameters from baseline 18F-FDG-PET/CT predict the development of distant metastatic disease?

Authors: R Hinzpeter, S Mirshahvalad, R Kulanthaivelu, A Kohan, C Ortega, U Metser, Z Liu, A Farag, E Elimova, R Wong, J Yeung, R Jang, P Veit-Haibach

Manuscript ID: Diagnostics-3015218

Specific Comments:

1. This study has various limitations, foremost among them being its retrospective nature and lack of external validation, as acknowledged by the authors. As such, the proposed predictive models may have limited utility, if any, in broader clinical settings.

2. Given a patient population from a single center and a largely homogeneous imaging protocol, it's anticipated that there will likely be variations in the performance of the proposed models when applied to other datasets. To thoroughly evaluate the models' performance and address any concerns about the study design, external validation is highly recommended.

3. Further elaboration is needed on the methodology used for extracting radiomics features. Given the numerous variables involved in this process, such as the quantization algorithm, voxel interpolation scheme, and criteria for volume inclusion, among many others, radiomics analysis can be complex and may impact the reported results.

4. It seems that PET radiomics analysis utilized raw image intensity values instead of SUVs, as recommended by numerous prior studies. This needs to be rectified to ensure comparability of radiomics parameters across patients.

5. The tumor volume of the study populations need to be disclosed. Given the well-established volume dependence of most studied radiomics parameters, it is advisable to assess the performance of the proposed models in relation to tumor volume to determine if any correlation or dependency exists.

6. The selection of k-folded light gradient boosting machine (LGBM) as the classifier appears to be ad-hoc, considering the wide range of binary classifiers available. It is strongly recommended to provide clarification on the rationale behind this selection.

7. The statistical analysis that the authors took is problematic. Given that the majority of the statistical difference tests were conducted within a setting where multiple comparisons arise, p-values for significance should be adjusted accordingly.

8. In addition, performance comparison of the various proposed models needs to be more rigorous. The authors are recommended to apply the DeLong tests or similar to assess the discriminative ability of the developed classifiers.

9. A number of previous studies have shown the impacts on PET radiomics features and characteristics due to scanner variability on acquisition parameters and post-process variables (PMID: 34575619; PMID: 26251842; PMID: 31240330; PMID: 31941949). Although it may be beyond the scope of this study to investigate how these factors impact the performance of the proposed models, a discussion is certainly warranted.

10. The paper has typos, and the writing style is not professional enough. The manuscript could be further improved by senior professionals editing the language for style and grammar.

Comments on the Quality of English Language

See comments for the authors.

Author Response

Distinguished Reviewer #1

Specific Comments:

  1. This study has various limitations, foremost among them being its retrospective nature and lack of external validation, as acknowledged by the authors. As such, the proposed predictive models may have limited utility, if any, in broader clinical settings.

A: Many thanks for your accurate comment. We completely agree that the retrospective nature of our investigation and the lack of external validation is a drawback, which is, however, found in many other publications within the field as well. Also, one needs to look for initial data first retrospectively to establish a hypothesis before any prospective trial can be set up. One could view this current investigation as a pilot study to evaluate if there is any measurable signal to further look into. If the reviewer feels this should reflected in the title, we are happy to add that. We would also argue that we evaluated an above-average number of patients, which enabled at least a robust internal validation. We are, however, currently working on external validation. But since there are very few public/downloadable registry data available for oesophageal cancer imaged with PET/CT, we have to work on data-sharing agreements with cooperation partners, which take, due to the different nature of privacy regulations in different jurisdictions, a lot more time than one could reasonably assume.

  1. Given a patient population from a single center and a largely homogeneous imaging protocol, it's anticipated that there will likely be variations in the performance of the proposed models when applied to other datasets. To thoroughly evaluate the models' performance and address any concerns about the study design, external validation is highly recommended.

A: We again completely agree with your standpoint and will continue our work on external validation as the next step in our project research pipeline. We also added your accurate statement to the conclusion to emphasize this fact.

  1. Further elaboration is needed on the methodology used for extracting radiomics features. Given the numerous variables involved in this process, such as the quantization algorithm, voxel interpolation scheme, and criteria for volume inclusion, among many others, radiomics analysis can be complex and may impact the reported results.

A: Thanks for your comment. We have used one of the most commonly used, well-established protocols and software for radiomics data extraction. We followed a robust and previously validated methodology in terms of extraction, which is guided by the software and described. It has been described in several of our previous publications in the same way. There was no further manipulation of the images i.e. no quantization or voxel manipulation in any way. We generally do not believe in this type of data ‘preparation’ for radiomics analysis. While it has been shown to generate better results for radiomics analysis, we are of the opinion that such preparation has very little chance of broad-based clinical implementation. Statistics and model building were done by a professional expert statistician in this field. We have added some additional comments in the model building section and hope that this is now clearer.

  1. It seems that PET radiomics analysis utilized raw image intensity values instead of SUVs, as recommended by numerous prior studies. This needs to be rectified to ensure comparability of radiomics parameters across patients.

A: Thanks for raising this point. As you accurately mentioned, SUVs are too crucial and well-known parameters to be overlooked. Our feature extraction included all SUV parameters (max, min, mean, and peak) to follow the most robust all-included evaluation possible. We clarified this in MM.

  1. The tumor volume of the study populations need to be disclosed. Given the well-established volume dependence of most studied radiomics parameters, it is advisable to assess the performance of the proposed models in relation to tumor volume to determine if any correlation or dependency exists.

A: We are not sure we understood the comment precisely. We included a wide range of parameters, and in fact, tumour volume stood out from the variables, being one of the two significant radiomics parameters in the multivariate analysis, retaining its significance even alongside prominent clinical factors (age, M1 disease). Thus, we think we already incorporated the tumour volume in our analyses and showed its respective independence in the multivariate analysis. Please let us know if this is meant or if the reviewer refers to any other tumour volume evaluation.

  1. The selection of k-folded light gradient boosting machine (LGBM) as the classifier appears to be ad-hoc, considering the wide range of binary classifiers available. It is strongly recommended to provide clarification on the rationale behind this selection.

A: Many thanks for the raised point. We added a clarification in this regard to the MM.

  1. The statistical analysis that the authors took is problematic. Given that the majority of the statistical difference tests were conducted within a setting where multiple comparisons arise, p-values for significance should be adjusted accordingly.

A: We are completely of the same opinion that the level of significance should be tailored to the purpose. We adjusted the level of significance when picking the variables for modelling. This is why we mentioned in MM that: “Although otherwise specified, a two-tailed p-value <0.05 was considered statistically significant.” We then specified in the results that we changed the level to <0.01 for the eligibility assessment of the radiomics features to enter modelling. This was done to enhance the model-building process and avoid any possible bias in this regard. This is copied here below for the reviewers’ convenience:

“In summary, 34 CT and 70 PET features differed significantly (p-values <0.01) between early-stage and advanced cohorts.”

  1. In addition, performance comparison of the various proposed models needs to be more rigorous. The authors are recommended to apply the DeLong tests or similar to assess the discriminative ability of the developed classifiers.

A: Many thanks for your valuable comment. As you suggested, we performed DeLong’s test for pairwise AUC comparison. However, DeLong’s test is not necessarily a more definite tool for the determination of a model's superiority than other tests. Thus, although we performed the comparison and reworded the text to be accurate based on your comment, we kept our final model since it had the most robust performance metrics overall. Another crucial thing in this regard is that the assumption of DeLong’s test is that the predictions used to generate the ROC curves are independent within pairs. However, shared covariates can violate this assumption, particularly in a scenario like ours, making this test’s results potentially less reliable. Our final model was selected (incorporating all parameters), because it overall showed the best metrics, though not having the statistically highest AUC. For completeness, however, we can add DeLong’s test result to the supplementary material if the reviewer/editor feels this enhances the understanding for readers.

  1. A number of previous studies have shown the impacts on PET radiomics features and characteristics due to scanner variability on acquisition parameters and post-process variables (PMID: 34575619; PMID: 26251842; PMID: 31240330; PMID: 31941949). Although it may be beyond the scope of this study to investigate how these factors impact the performance of the proposed models, a discussion is certainly warranted.

A: Many thanks for your valuable comments. Please see the comment above, where we discussed the variability/harmonisation of radiomics evaluation. We however added a comment to the limitations section.

  1. The paper has typos, and the writing style is not professional enough. The manuscript could be further improved by senior professionals editing the language for style and grammar.

A: We have reviewed the manuscript again through the process of the rebuttal and resolved the typos. Several of the authors are native native-speakers. We hope it now meets your satisfaction.

Reviewer 2 Report

Comments and Suggestions for Authors

Today, especially in radiology, standardising imaging to include AI in the diagnosis and evaluation of treatments in oncology is one of the most exciting and important tasks.

Understanding the fact that the image is not only a simple picture with dots in black, white and grey but with multiple types of information which could be exploited in any way. This is the aim of this article - to improve the prediction of the metastatic process in oesophagal cancer. 

One question - excellent number of patients, long period of time of recruiting. The acquisition protocol of PET, I supposed, had changed over time. Does this fact influence the results?

Author Response

Distinguished Reviewer #2

Today, especially in radiology, standardising imaging to include AI in the diagnosis and evaluation of treatments in oncology is one of the most exciting and important tasks.

Understanding the fact that the image is not only a simple picture with dots in black, white and grey but with multiple types of information which could be exploited in any way. This is the aim of this article - to improve the prediction of the metastatic process in oesophagal cancer. 

One question - excellent number of patients, long period of time of recruiting. The acquisition protocol of PET, I supposed, had changed over time. Does this fact influence the results?

A: Thanks for taking the time to review our paper and for your favourable comments.

Regarding your question, we limited the time span of our retrospective review to an interval with only one change in imaging devices. Thus, there were mostly homogeneous reconstruction protocols and administrated dosage (as much as possible). We have added the additional PET/CT and reconstruction parameters to the MM. Also, notably, we did not change/prepare any of our imaging data because we believe that clinical implementation would only occur when these methods work across different systems and acquisitions.

We also added a section to the limitation in this regard so that the readers become aware of the possible unknown impacts this issue can have on the findings.

Round 2

Reviewer 1 Report

Comments and Suggestions for Authors

Journal: Diagnostics

Manuscript Title: Gastro-esophageal cancer: Can radiomic parameters from baseline 18F-FDG-PET/CT predict the development of distant metastatic disease?

Authors: R Hinzpeter, S Mirshahvalad, R Kulanthaivelu, A Kohan, C Ortega, U Metser, Z Liu, A Farag, E Elimova, R Wong, J Yeung, R Jang, P Veit-Haibach

Manuscript ID: Diagnostics-3015218-R1

The authors have addressed many of the concerns from the earlier reviews, and I appreciate the efforts made to a) provide more details on the classification model, b) correct and improve the statistical analysis, and c) acknowledge major inherent weaknesses of the study. One comment left unaddressed is whether the radiomics parameters were extracted from the SUV maps (previous round comment #4). This needs to be clarified. If not done so already, the authors are advised to use SUV maps for feature calculation to ensure comparability with the rapidly growing body of PET-related radiomics studies.

Comments on the Quality of English Language

Minor editing of the English language may be needed.

Author Response

Dear Distinguished Reviewer,

We are happy that we could address other raised concerns. We believe your suggestions significantly enhanced our manuscript.

Thanks for raising this comment again to open the door for a discussion. We would like to clarify that the radiomics parameters were extracted from the corrected SUV images, not from raw data or non-corrected data. They were, however, not normalized to a reference tissue, and there was no evaluation on dedicated SUV maps. We would like to reiterate that the radiomics extraction was done in an IBSI-compliant way. SUV maps are possibly useful i.e., in brain imaging, where there is high uptake in an adjacent reference tissue. This is, however, not the case for gastro-esophageal cancer. The normal esophagus would be a less-than-ideal reference since there often can be, i.e., inflammation in the remaining esophagus, which then would be a confounding factor for the generation of the maps.

Ultimately, such measures are not recommended in SNMMI/EANM guidelines (https://doi.org/10.1007/s00259-022-06001-6).

Overall, as we would like to avoid criticism from the readership about not being IBSI-compliant or not being compliant with the latest guidelines, we do not think that generating new maps for the data extraction would be of great additional value. Having said that, we will investigate its value in future studies.

Also, please consider the limited time available for a revision (5 days).

Many thanks again for your precious comments and for helping us to significantly improve our study presentation.